# Marine ice sheet model performance depends on basal sliding physics and sub-shelf melting

Rupert Michael Gladstone[1,2,3], Roland Charles Warner[2], Benjamin Keith Galton-Fenzi[2,4], Olivier Gagliardini[5], Thomas Zwinger[6], and Ralf Greve[7]

[1]VAW, Eidgenössische Technische Hochschule Zürich, ETHZ, Switzerland
[2]Antarctic Climate and Ecosystems Cooperative Research Centre, University of Tasmania, Hobart, Australia
[3]Arctic Centre, University of Lapland, Rovaniemi, Finland
[4]Australian Antarctic Division, Kingston, Tasmania, Australia
[5]Univ. Grenoble Alpes, CNRS, IRD, IGE, F-38000 Grenoble, France
[6]CSC – IT Center for Science Ltd., Espoo, Finland
[7]Institute of Low Temperature Science, Hokkaido University, Japan

*Correspondence to:* Rupert Gladstone (RupertGladstone1972@gmail.com)

**Abstract.**

Computer models are necessary for understanding and predicting marine ice sheet behaviour. However, there is uncertainty over implementation of physical processes at the ice base, both for grounded and floating glacial ice. Here we implement several sliding relations in a marine ice sheet flowline model accounting for all stress components, and demonstrate that model

resolution requirements are strongly dependent on both the choice of basal sliding relation and the spatial distribution of ice shelf basal melting.

Sliding relations that reduce the magnitude of the step change in basal drag from grounded ice to floating ice (where basal drag is set to zero) show reduced dependence on resolution compared to a commonly used relation, in which basal drag is purely a power law function of basal ice velocity. Sliding relations in which basal drag goes smoothly to zero as the grounding

line is approached from inland (due to a physically motivated incorporation of effective pressure at the bed) provide further reduction in resolution dependence.

A similar issue is found with the imposition of basal melt under the floating part of the ice shelf: melt parameterisations that reduce the abruptness of change in basal melting from grounded ice (where basal melt is set to zero) to floating ice provide improved convergence with resolution compared to parameterisations in which high melt occurs adjacent to the grounding line.

Thus physical processes, such as sub-glacial outflow (which could cause high melt near the grounding line), impact on capability to simulate marine ice sheets. If there exists an abrupt change across the grounding line in either basal drag or basal melting then high resolution will be required to solve the problem. However, the plausible combination of a physical dependency of basal drag on effective pressure, and the possibility of low ice shelf basal melt rates next to the grounding line, may mean that some marine ice sheet systems can be reliably simulated at a coarser resolution than currently thought necessary.

# 1 Introduction

Ice Sheet Models (ISMs) are increasingly being used in process studies, sensitivity studies and projections of Marine Ice Sheet (MIS) future behaviour (Joughin et al., 2010; Favier et al., 2014; Gong et al., 2014), and Model Intercomparison Projects (MIPs) to investigate the ice sheet response to ocean forced basal melting of ice shelves are currently in their design phase (Asay-Davis et al., 2016).

Past ISM studies have shown inconsistent grounding line behaviour at typical resolutions (Vieli and Payne, 2005). Inconsistencies were very large (grounding line discrepancies of $\approx$100 km) for grid resolutions of $\approx$10 km, and typically still not converged for grid resolutions of $\approx$1 km. Studies in which simulations are carried out at multiple different mesh resolutions usually demonstrate convergent behaviour, but very fine resolution is often needed to approach a converged solution (Cornford et al., 2013; Gladstone et al., 2010a; Pattyn et al., 2012). Practical solutions have been suggested, such as parameterising the flux of ice across the grounding line as a function of ice thickness (Schoof, 2007; Pollard and DeConto, 2009), parameterising the grounding line position at sub-grid resolution (Gladstone et al., 2010b; Seroussi et al., 2014), or implementing adaptive mesh refinement to provide very high resolution at and near the grounding line (Cornford et al., 2013; Durand et al., 2009). These solutions all have limitations, and the computational cost of running a sufficiently high resolution ISM to robustly represent grounding line motion remains high, even with adaptive refinement.

However, model-based MIS studies (e.g. Pattyn et al. (2012)) typically use a simple basal traction prescription, or "sliding relation" (Weertman, 1957), which neglects the impact of effective pressure at the bed (or equivalently "height above bouyancy") on basal shear stress. The inclusion of pressure dependency (reviewed by Fowler (2010)) provides a physical motivation for smoothing out what is otherwise a large step change in basal drag across the grounding line. This was proposed over 30 years ago (e.g. Budd et al. (1984)) and may affect the resolution requirements for successful grounding line modelling. Recent treatments of basal traction that also vanish smoothly at the grounding line include Leguy et al. (2014) and Tsai et al. (2015).

Furthermore, the implications of imposing basal melting (note that in the current study "basal melting" refers always to melting under the ice shelf and not the grounded part of the MIS) on resolution requirements have not been explicitly investigated. In the current study, we assess the impact of choosing between different sliding relations, and between different approaches to parameterising basal melting, on model resolution requirements in a Stokes flow ice dynamic model.

# 2 Methods

We use the ice dynamic model Elmer/Ice (Gagliardini et al., 2013). The Stokes equations for a viscous fluid with non-linear rheology are solved using the finite element method over a two-dimensional flowline domain (one vertical and one horizontal dimension) in which lateral drag is parameterised (Gagliardini et al., 2010) according to channel width, $W$, and a contact problem is solved to determine the evolving grounding line position (Favier et al., 2012).

The rheology follows Glen's law (Glen, 1952; Paterson, 1994) with viscosity calculated using a temperature dependent Arrhenius law (Gagliardini et al., 2013; Paterson, 1994). Temperature is held constant at -10 C for all simulations in the current study.

We implement alternative sliding relations (Section 2.1) and a basal melt parameterisation (Section 2.2) in Elmer/Ice.

## 2.1 Basal sliding

The form of the sliding laws used in the current study is motivated by early laboratory sliding experiments (Budd et al., 1979) and Antarctic simulations (Budd et al., 1984), which suggested modifying the original Weertman sliding relation (Weertman, 1957) by incorporating a power-law dependence of the drag on effective pressure at the bed as follows:

$$\tau_b^p = -Cu_b^m z_*^q, \tag{1}$$

where $\tau_b$ is basal shear stress, $u_b$ is basal ice velocity, $z_*$ is the height above buoyancy (related to effective pressure at the bed, $N$, by $N = \rho_i g z_*$), $m$, $p$ and $q$ are constant exponents, and $C$ is a constant sliding coefficient. Besides the laboratory studies of ice sliding already mentioned, the introduction of basal effective pressure into sliding in the 1980s, particularly in the context of a marine ice sheet and the identification (via $z_*$) with ocean pressure, was further motivated by a characteristic feature of West Antarctica's fast flowing ice streams: increasingly rapid flow towards the grounding line (and generally decreasing $z_*$) despite

a steadily decreasing surface slope and hence gravitational driving stress. Various parameterisations were developed from the information about velocities, surface slopes and ice thicknesses then available (see e.g. Budd et al. (1984) and references therein).

In the current study we set $m = \frac{1}{3}$ and $p = q = 1$ for all simulations. These values for $p$ and $q$ are chosen for simplicity, and deviate from the original values tuned for large scale ice sheet simulations (Budd et al., 1984). We impose $z_* \geq 0$ when

calculating $\tau_b$. Ideally $z_*$ would be calculated using basal water pressure from a sub-glacial hydrology model. In the current study, we simply use hydrostatic balance based on sea level,

$$z_* = \begin{cases} H, & \text{if } b >= 0 \\ H + b\frac{\rho_o}{\rho_i} & \text{if } b < 0 \end{cases} \tag{2}$$

where $H$ is local ice thickness, $b$ is the bedrock elevation relative to sea level (positive upwards), $\rho_o$ is the density of ocean water, and $\rho_i$ is the density of ice. This is equivalent to assuming a sub-glacial hydrology system fully connected to the ocean.

The four sliding relations used in the current study are given by

$$\tau_b = -C_1 u_b^{\frac{1}{3}}, \tag{3}$$

$$\tau_b = -C_2 u_b^{\frac{1}{3}} z_*, \tag{4}$$

$$\tau_b = -C_3 u_b^{\frac{1}{3}} \frac{z_*}{H}, \tag{5}$$

$$\tau_b = -C_4 u_b^{\frac{1}{3}} (z_* + z_o), \tag{6}$$

where $z_o$ is a thickness offset and $C_n$ are sliding coefficients. The first two sliding relations (given by equations (3) and (4), and henceforth referred to as SR1 and SR2 respectively) are specific cases of equation (1) and derive from previously published sliding laws. SR1 is widely used in model intercomparison studies, such as the Marine Ice Sheet Model Intercomparison Project (MISMIP, Pattyn et al. (2012)), and features an abrupt change in basal shear stress from grounded to floating ice. It is commonly referred to as "Weertman sliding" after Weertman (1957). SR2 implements a smooth transition of basal drag to zero as the grounding line is approached from landwards.

The form of SR2 is based on modifying SR1 for our study. It is motivated by parameterisations (McInnes and Budd, 1984) of sliding relations for fast flowing ice streams in West Antarctica, where a linear relation between $\tau$ and $z_*$ was observed to hold towards the grounding line, although their parameter choice was $p = q = m = 1$. It is worth noting that a number of other sliding relations have been published in which the transition of basal shear stress across the grounding line is less abrupt that in SR1. For example (Pattyn et al., 2006) implemented a fixed size transition zone. Theoretical work for the case of sliding with cavitation (Schoof, 2005; Gagliardini et al., 2007) has also been used to develop sliding relations in models (Leguy et al., 2014; Tsai et al., 2015), though it is not clear whether the assumptions made are applicable in all real world cases of glacier sliding. The current study does not aim to promote use of any particular sliding relation, but rather to explore a specific aspect of the sliding implementation, namely the abruptness with which basal shear stress goes to zero as the grounding line is approached. Sliding relations SR3 and SR4 implement further modifications to SR2 in order to explore this aspect of sliding. SR3 (equation 5) uses thickness scaling to give a law which captures the smooth fade to zero of basal drag approaching the grounding line of SR2, but which equates to the familiar SR1 for ice grounded above sea level, providing ice sheet profiles more directly comparable to SR1. SR3 can be regarded as restricting the assumption that basal water pressures is directly tied to ocean pressures as one moves inland from the grounding line. The aim of SR4 (equation 6) is to provide a step-change in basal drag from grounded to floating ice, but one with significantly smaller magnitude than would occur with a Weertman-type (SR1) sliding relation. The sliding relations and their coefficient values are summarised in Table 1. The coefficient values were chosen to give approximately similar grounding line positions after the initial spin up and advance experiments.

## 2.2 Ice shelf basal melt

We implement a parameterisation for ice shelf basal melt rate, $m_b$, similar to that used in the "Marine Ice Sheet Ocean Model Intercomparison Project" phase 1 (MISOMIP1), and described in the MISOMIP1 experimental setup (Asay-Davis et al., 2016). This parameterisation in its original form increases with depth due to the pressure-driven depression of the sea water freezing point along the ice shelf base, and hence would generally give a maximum in $m_b$ adjacent to the grounding line. However, a parameterisation that only considers the pressure-enhanced "thermal driving" does not account for the sub-ice cavity geometry that may limit oceanic heat transport right to the grounding line. Nor does it account for the impact of a sub-glacial ouflow that may trigger or strengthen a bouyant meltwater plume at the ice-ocean interface.

Ice shelf melt rates close to (within 20-30 km of) the grounding lines of major Antarctic outlet glaciers are typically much higher than ice shelf average values, sometimes by an order of magnitude (Rignot and Jacobs, 2002). However, Rignot and Jacobs (2002) did not investigate the spatial patterns of melt rates within the regions close to the grounding line. A plume model

study (Jenkins, 2011) suggests that, in the presence of sub-glacial outflow at the grounding line, significant melting can occur adjacent to the grounding line, and that the maximum in melting is likely to occur close to, but not adjacent to, the grounding line. A 3D ocean modelling study (Galton-Fenzi, 2009) is in agreement with this result, and further indicates that a reduction in strength of sub-glacial outflow can reduce the strength of melting adjacent to the grounding line. Simulations using a plume model with no sub-glacial outflow (Parizek and Walker, 2010) show a melt rate that peaks tens of km from the grounding line and decays to zero at the grounding line.

Here we implement an optional melt scaling parameter, $S_w$, used to reduce $m_b$ smoothly to zero as the grounding line is approached from the ice shelf. By carrying out simulations both with and without this melt scaling we effectively implement two opposite end-members for the melt distribution: a smooth transition to zero melting at the grounding line and maximum melting adjacent to the grounding line.

The melt rate $m_b$ is calculated in m a$^{-1}$ ice equivalent and is parameterised as a function of depth by

$$m_b = S_w S_i \frac{c_w \gamma_T}{L} \Omega \Delta T, \tag{7}$$

where $\Delta T$ is the "thermal driving", $L$ is the latent heat of fusion of ice, $c_w$ is the heat capacity of seawater, $\gamma_T = 10^{-4}$ is a heat transfer coefficient, $\Omega$ is a dimensionless tuning parameter, and $S_w$ and $S_i$ are scaling factors. The thermal driving is the far field to local temperature difference, $\Delta T = T_f - T_o$, where $T_f$ is the local freezing point of sea water, and $T_o$ is the far field ocean temperature. $T_f$ is approximated here in degrees Celsius using $T_f = -1.85 + 7.61 \times 10^{-4} z_i$, where $z_i$ is the depth of the ice base relative to sea level (positive upwards). We set $T_o = 2.0$ C for our simulations.

The scaling factor $S_w$ is implemented as a function of water column thickness, $H_w$ (given by $H_w = z_i - b$), to reflect the influence of cavity geometry,

$$S_w = \tanh\left(e\frac{H_w}{H_{w0}}\right), \tag{8}$$

where $H_{w0}$ is a reference water column thickness. $S$ approaches 1 in deeper water ($H_w > H_{w0}$). We present simulations both with and without the water column thickness scaling. Where it is used we set $H_{w0} = 100$ m. Where it is not used we set $S_w = 1$.

Iceberg calving is not represented in the current study, and the ice shelf front position remains fixed. This results in a vanishingly thin ice shelf in some simulations and can cause numerical instabilities and model failures. $S_i$ is an ice shelf depth scaling parameter introduced to avoid the occurrence of a vanishingly thin ice shelf. $S_i$ is given by

$$S_i = \max\left[\tanh\left(e\frac{z_{i0} - z_i}{z_s}\right), 0\right], \tag{9}$$

where $z_{i0}$ is a reference ice base height relative to sea level (positive upwards) and $z_s$ is a (directionless) scaling depth. In practice the use of $S_i$ gives zero melting for $z_i > z_{i0}$. The $S_i$ scaling is used in all simulations with values $z_{i0} = -40$ m and $z_s = 100$ m.

In the simulations presented here melt is applied to all mesh nodes in the floating part of the ice sheet. Simulations were also carried out in which melting was also applied to the last grounded node, and this was found not to cause a large difference: the results and interpretation presented here hold for both cases. The experiments are described in Section 2.3 and Table 3.

## 2.3 Experiment design

The experimental set up involves an 1800 km domain with linear down sloping bedrock, $b$, given in km relative to sea level by

$$b = 0.2 - \frac{0.9x}{1800}, \tag{10}$$

where $x$ is horizontal distance in km from the ice divide. This gives a bed rock elevation varying between $z = 200$ m and $z = -700$ m, where $z$ is the vertical coordinate measured relative to sea level.

Net surface accumulation, $a$, is given in m a$^{-1}$ by

$$a = \frac{x}{1800} \frac{\rho_o}{\rho_i}. \tag{11}$$

The upstream boundary represents the ice divide, and a Dirichlet condition is used here to set the horizontal component of velocity to zero, ensuring flow symmetry. An external hydrostatic pressure distribution imposed by the ocean (below sea level) is prescribed at the spatially fixed downstream calving front. This external pressure is also applied to the base of the ice shelf.

The mesh is composed of quadrilateral elements with 11 equally spaced layers in the vertical direction. Each experiment has been run three times with different resolutions in the horizontal (Table 2). The resolutions chosen are indicative of resolutions that could be achieved by large scale Stokes simulations of ice sheets with the current generation of models. Thus they are coarser than is commonly considered to be required for self-consistent simulations involving grounding line movement (Durand et al., 2009; Pattyn et al., 2012). This is intentional, so that the current study may assess the potential for different sliding laws or basal melt parameterisations to achieve resolution-independent behaviour at coarser resolutions than is required with sliding relations similar to SR1.

The experiments are summarised in Table 3. Spin-up is performed in two stages: The first stage ("SPIN", Table 3) is from a uniform thickness (300 m) slab of ice for 40 ka with parameterised channel width 1000 km (very low buttressing). The second stage ("ADVA", Table 3) constitutes a further 40 ka with parameterised channel width 150 km (significant buttressing). This two-stage spin-up is carried out separately for each resolution and sliding law. The purpose of including buttressing is to provide a mechanism for basal melting under the shelf to impact on grounded ice. This impact is through ice shelf thickness change: a thicker ice shelf provides more buttressing. Note that basal melting is zero during both stages of the spin-up.

Retreat simulations are then carried out (experiment names beginning with "R" in Table 3), which form the main focus for this study. The cause of retreat is a change in forcing. In the "Retreat due to Butressing reduction" experiment (RB), the forcing change is a reduction in the lateral drag back to a parameterised channel width of 1000 km. In the "Retreat due to High melt with Water column scaling" experiment (RHW) basal melting (Section 2.2) is imposed under the ice shelf. Both forcing changes are applied together in the "Retreat due to High melt with Water column scaling and Buttressing reduction" experiment (RHWB). For the melt induced retreat simulations we set $\Omega$ to 0.045 or 0.009 (Table 3), resulting in typical melt rates between 1 and 10 m a$^{-1}$. A variation on RHW is the "Retreat with Low melt with Water column scaling" experiment (RLW).

We also carry out re-advance experiments (ALW and AL in Table 3) to test whether simulations reach the same steady state in advance as in retreat under identical forcing. The full set of experiments and corresponding parameters are given in Table 3.

Individual simulations are referred to in the results section by their "simulation code", made up of the experiment name (Table 3), the sliding relation used (Table 1) and the resolution (Table 2). For example, SPIN_SR1_R1 is the initial spin up with Weertman sliding and an element size of 3.6 km.

## 3 Results

Our main criterion for assessing the results is resolution dependency. The model used here, Elmer/Ice, has been demonstrated in previous studies (e.g. Durand et al. (2009); Gagliardini et al. (2013, 2016) to give convergent behaviour with resolution, i.e. its output approaches a self-consistent solution as resolution is made increasingly fine. In the current study we do not attempt to demonstrate convergence in all cases (indeed convergence is certainly not achieved in all cases), but instead consider the dependence on resolution across the three resolutions used (Table 2), under the premise that weaker dependence on resolution is an indicator of being closer to the converged solution. The causes of strong resolution dependency in the current study will be discussed in Section 4.

Specifically, we consider experiments in which the grounding line position differs between simulations of different resolution by distances of approximately the same magnitude as the size of a single element not to have significant dependency on resolution. Conversely, we consider experiments with grounding line differences of several element sizes or greater to have significant dependence on resolution. For example, differences in grounding line position of the order of 100 km between simulations at different resolutions are considered to indicate significant resolution dependency, whereas differences of the order of 1 km are not. Similarly, when we say "near to the grounding line" we are also talking in terms of element size. For example "high melt near the grounding line" can be interpreted as "high melt within a very small number of elements of the grounding line".

We focus mainly on the evolution of grounding line position. The spinup simulations (SPIN and ADV, Table 3) do not vary significantly with resolution, and so our analysis focuses on retreat and re-advance simulations. The fact that the spinup simulations show very little dependency on resolution is not an indicator that the retreat simulations should show equally low dependency on resolution. Previous studies have shown that ice sheet models often demonstrate much higher resolution dependency in retreat than advance, or vice versa (Pattyn et al., 2012; Gladstone et al., 2012). This discrepancy between retreat and advance experiments implies multiple possible steady states for a given forcing (Gladstone et al., 2010a).

However, the ice geometry in the spinup simulations does vary significantly with choice of sliding relation (Figure 1). The steady state ADVA_SR1 profiles have their steepest surface slope close to the grounding line due to the step change in basal drag from grounded to floating ice (Figures 1 and 2). The steady state SR2 and SR4 profiles (Figure 1) have their greatest surface slope further inland where the overburden pressure becomes important. The steady state SR3 profile is similar to SR1 towards the ice sheet interior, but is thinner due to having lower drag close to the grounding line. It is very similar to SR2 near the grounding line.

Figure 2 also shows the shear stress component of the Cauchy stress tensor, $\sigma_{xz}$, for SR1 and SR3. Given the low gradient of the bed, $\sigma_{xz}$ at the bed is an approximation to basal drag. Shear stress peaks at the grounding line for SR1 due to the step

change in basal drag. The inclusion of dependence on effective pressure at the bed via $z_*$ in SR3 leads to a gradual change and a much larger transition zone.

The impact of choice of sliding relation on the way the modelled ice sheet responds to changing resolution is shown for retreat simulations in Figure 3. The sliding relations featuring a step change in basal drag across the grounding line (SR1,

SR4a and SR4b, shown in teal, black and blue respectively - the top three families of curves in the upper plot) do not exhibit consistent behaviour with resolution, with the RB_SR1_R0 simulation in particular showing no retreat of the grounding line after the buttressing reduction. Note that of these three simulations the magnitude of this basal drag step change is smallest for SR4b and largest for SR1. The results in Figure 3 are consistent with a smaller step change in basal drag being indicative of better convergence with resolution, similar to a previous result when using a "shelfy-stream" ice sheet model (Gladstone

et al., 2012). However, even SR4b still shows significant resolution dependency, indicating that much finer resolution than is considered here would be required for a reliable simulation. The sliding relations in which basal drag goes smoothly to zero as the grounding line is approached (SR2 and SR3, shown in green and red respectively) show the most consistent behaviour with resolution in experiments RB and RHW.

The purpose of the RHW experiment (Figure 3, lower panel), as distinct from the RB experiment, is to test dependence on

resolution in the presence of basal melting. Because SR1 and SR4a showed strong resolution dependence already in RB they have been omitted from the RHW experiments. In general the consistency across different resolutions appears weaker in the case of the melt-induced retreat simulations (RHW) than the reduced buttressing simulations (RB). The ice sheet profiles at the end of the RHW simulations are outlined in grey in Figure 1 for sliding relations SR2, SR3 and SR4b.

The SR3 retreat simulations are unique in exhibiting an overshoot: after a strong initial grounding line retreat a small

advance is seen. In the RHW_SR3_R2 simulation damped oscillations can be seen. The reason for this behaviour is not clear, but the lower resolution simulations fail to exhibit this behaviour, indicating at least some resolution dependency in this experiment. The grounding line positions in the RHW_SR3 simulations also do not show a monotonic progression with increasing resolution.

Figure 4 shows more clearly the resolution dependency in the final grounding line positions for the RB and RHW experi-

ments, after 40 ka. Again, R1, R4a and R4b show much stronger resolution dependence than R2 and R3.

We now look more closely at the impact of basal melting on resolution dependency. We compare retreat and re-advance simulations. These experiments involve the lower melt and greater buttressing scenarios, applied to different starting configurations (see Table 3). We also investigate the impact of the water column scaling, in which zero melt is approached close to the grounding line. Sliding relation SR2 is used for these experiments as it has shown much weaker resolution dependency than

SR1 and SR4, and has not shown difficult-to-interpret behaviour such as the damped oscillations in RHW_SR3_R2.

Figure 5 shows both retreat (RL and RLW, red lines) and advance (AL and ALW, black lines) simulations with water column scaling either on (RLW and ALW) or off (RL and AL). The advance simulations have identical inputs to the corresponding retreat simulations in all respects except for initial conditions. Note that while the majority of these simulations were run for 20 ka, the AL simulations were run for 40 ka because 20 ka was not long enough to approach a steady state.

In the presence of water column scaling the advance and retreat simulations approach the same grounding line position at all resolutions, showing no significant resolution dependency (right hand plots of Figure 5). This is consistent with the premise that a unique solution exists, which might be expected behaviour on a linear down sloping bed (Schoof, 2007), although this has not been proven in the presence of buttressing and basal melting that depend on ice shelf geometry.

However, where there is a large step change in basal melt across the grounding line (AL and RL, left hand plots of Figure 5), the advance and retreat grounding lines do not approach the same final position. Behaviour is strongly resolution dependent, especially in the re-advance experiment. It is unclear whether retreat and re-advance simulations would eventually converge to the same solution, and finer resolution simulations would be required to determine this with confidence.

Dependency on resolution appears to be stronger in the case of advance experiments than retreat experiments. This is in sharp contrast to the SPIN and ADVA experiments, which are a kind of advance experiment (in that the grounding line position is advancing through the simulation toward its final position), in which no significant resolution dependency was observed. This suggests that it is specifically the melting which causes resolution dependence, and that it causes greater resolution dependence in advance than in retreat.

The step changes in grounding line position during the early stages of retreat (Figure 5 upper panels) are typically indicative of a single element retreat for RLW_SR2, but are typically multiple element retreat steps in RL_SR2.

## 4   Discussion

As in previous studies with different ice dynamic models (e.g. Pattyn et al. (2006); Durand et al. (2009); Gladstone et al. (2012)), a step change in basal drag across the grounding line causes strongly resolution dependent behaviour in the current study using the Elmer/Ice finite element Stokes flow model. A large step change causes stronger resolution dependency than a smaller step change. A comparable resolution dependency on basal melt is shown in the current study: a step change in basal melt across the grounding line causes significant resolution dependent behaviour, worse for larger step changes. Cases demonstrating strong resolution dependence at the resolutions presented here are of low interest to the current project, which aims to identify situations where such resolutions are viable. Much weaker resolution dependence is found in the current study in the case where both basal drag and basal melt approach zero as the grounding line is approached from landward and seaward respectively.

A change in value of sliding coefficient for a given sliding relation can also impact on resolution requirements (Gladstone et al., 2012). But since SR1 will typically give a global maximum basal shear stress at the grounding line, and SR2 will typically give a global minimum basal shear stress at the grounding line, it is expected (and this is borne out by comparing Gladstone et al. (2012) to the current study) that choice of sliding relation has much greater impact on resolution requirements than the magnitude of the sliding coefficient.

The results of the melting experiments have important implications for application of model studies to real marine ice sheet systems. We have shown that even when the ice sliding relation permits resolution independent simulations at the widely achievable resolutions used in the current study, this situation can be negated by the abruptness of spatial onset of ice shelf

basal melting. In RHW_SR2, RLW_SR2 and ALW_SR2 experiments, where the onset of basal melting was gradual due to the scaling factor $S_w$ (equation 8), acceptable behaviour was observed over the sequence of resolutions we explored. However, even in low melt rate scenarios, the absence of this gradual transition gave rise to much more significant resolution dependence and a failure of retreat and readvance simulations to arrive at a unique grounding line location. Clearly more studies are required to explore the influence of abrupt spatial onset of melting. As discussed earlier (Section 2.2) high melt rates are observed within tens of km of the grounding lines of major Antarctic outlet glaciers, with the likelihood that such melt rates occur immediately adjacent to the grounding line in the presence of strong sub-glacial outflows. Accordingly, marine ice sheet systems with low surface slopes near the grounding line (indicating low basal drag approaching the grounding line) and with low basal melting near the grounding line (such as might be the case in the absence of strong sub-glacial outflow) would likely be more easily achievable targets for modelling studies at the resolutions explored in the present study. For model studies of less tractable systems, very high resolution would be needed near the grounding line. While sub-grid parameterisations for grounding line position or cross-grounding line ice flux have been developed (e.g. Pattyn et al. (2006); Pollard and DeConto (2009); Gladstone et al. (2010b); Feldmann et al. (2014)), there is clearly a new challenge to handle the influence of onset of basal melting on the near grounding line dynamics. Furthermore, parameterisations that work well in the absence of ice shelf basal melting will need to be tested in the presence of melting, and may need to be modified.

The basal melt parameterisations presented here, in particular the choice of whether or not to implement water column scaling, are intended to provide opposing end members in terms of melt distribution near the grounding line. The current study has demonstrated the impact this choice has on required model resolution, but does not advocate a particular melt parameterisation. Similarly, the sliding laws SR1 and SR2 are opposing end members in terms of basal shear stress near the grounding line. The choice of sliding relation has been shown to impact on resolution requirements, but a specific sliding relation is not advocated here. The choice of both melt parameterisation and sliding relation should be governed by the physical processes, not by numerical convenience. Our aim has been to demonstrate that different physical systems can have different resolution sensitivities.

The sliding relations presented here in which dependence on effective pressure at the bed is incorporated would have a stronger physical justification if used in conjunction with a computer model for sub-glacial hydrology, to replace the assumption that the hydrologic system is everywhere in contact with the ocean with a physically justifiable effective pressure distribution. It might be expected that, for the case of efficient channelised sub-glacial drainage (Hewitt et al., 2012; Werder et al., 2013), a strong hydrologic connection to the ocean may exist. However, in such a case, there may be very high local variations in basal water pressure, resulting in sticky spots (relatively low basal water pressure and hence high basal shear stress) in between active channels. To simulate such a system a very high model resolution would need to be used to represent basal processes, and potentially also for the grounding line, if these sticky spots are present close to the grounding line. For the case of less efficient "distributed" drainage (Schoof et al., 2012; Werder et al., 2013), a lower resolution would suffice for the hydrology system, and perhaps also for the grounding line, since there would likely be uniformly high basal water pressures (i.e. low effective pressure) near the grounding line. Studies of grounding line behaviour in a coupled hydrology-ice sheet model would be of great benefit to further understand this issue.

The results from the current study appear to be in conflict with the findings of Gagliardini et al. (2016), who found that imposing a fixed length transition zone near the grounding line (similar to that proposed by Pattyn et al. (2006)), where the basal drag is scaled linearly to zero as the grounding line is approached (from landward), did not significantly reduce the resolution requirements. There are, however, a number of significant differences between the current study and Gagliardini et al. (2016), such as the use of a direct physical motivation to impose the drag reduction in the current study, rather than imposition of linearity. We speculate that the key factor is that the imposed linear transition zone of Gagliardini et al. (2016) is typically of the same order of magnitude as the element size, meaning that the step change in basal drag across the grounding line, while moderately reduced, is not reduced by an order of magnitude or more, as in the current study for SR2 and SR3. The effect of incorporating dependence on basal effective pressure on the basal stress gradient approaching the grounding line is evident for the current study in Figure 2. The transition zone is several hundred km for SR3. A future study with further simulations will be needed to fill the gap in experiment design between the two studies to confirm whether this difference in transition zone size is the actual explanation for the differences in resolution dependence between the two studies.

## 5 Conclusions

We have demonstrated that resolution requirements for marine ice sheet simulations with an evolving grounding line are highly sensitive to the physical implementation of both basal sliding and ice shelf basal melting. In particular a large step change in either basal drag or basal melting across the grounding line can cause strong dependence of model behaviour on resolution.

Any marine ice sheet modelling studies whose outcomes involve a moving grounding line should demonstrate convergent behaviour with resolution over the region of parameter space relevant to their experimental setup, bearing in mind that basal drag and basal melt can both cause resolution dependence, and that resolution dependence may differ for an advancing and a retreating grounding line.

A significant implication of the current study is that conducting transient Stokes flow simulations of whole marine ice sheets, such as century scale simulations of the West Antarctic Ice Sheet for example, is a potentially tractable problem where evidence supports both basal drag and basal melting decreasing smoothly to zero as the grounding line is approached from respectively grounded and floating regions. Conversely, if there is a sharp onset of basal melting immediately beyond the grounding line, high resolution might be required regardless of the character of the basal sliding relation.

*Acknowledgements.* The authors wish to thank Stephen Cornford and Bill Budd for useful discussions about the simulations. The authors wish to acknowledge CSC - IT Centre for Science, Finland for computational resources. This research utilised the NCI National Facility in Canberra, Australia, which is supported by the Australian Commonwealth Government. Rupert Gladstone was funded from the European Union Seventh Framework Programme (FP7/2007-2013) under grant agreement number 299035. This research was supported in part by Academy of Finland grant number 286587. Ralf Greve was supported by a UTAS (University of Tasmania) Visiting Fellowship (September – December 2014), and by a JSPS (Japan Society for the Promotion of Science) Grant-in-Aid for Scientific Research A (No. 25241005).

This work was supported in part by the Australian Government's Cooperative Research Centres Programme through the Antarctic Climate and Ecosystems Cooperative Research Centre (ACE CRC).

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

**Table 1.** Sliding relations and constants used in the current study.

| Sliding Relation | Equation | Sliding coefficients, $C_n$ | $z_o$ |
|---|---|---|---|
| SR1 | 3 | $10^{-3}$ MPa m$^{-\frac{1}{3}}$ a$^{\frac{1}{3}}$ | - |
| SR2 | 4 | $7 \times 10^{-6}$ MPa m$^{-\frac{4}{3}}$ a$^{\frac{1}{3}}$ | - |
| SR3 | 5 | $10^{-3}$ MPa m$^{-\frac{1}{3}}$ a$^{\frac{1}{3}}$ | - |
| SR4a | 6 | $4 \times 10^{-6}$ MPa m$^{-\frac{4}{3}}$ a$^{\frac{1}{3}}$ | 100 m |
| SR4b | 6 | $4 \times 10^{-6}$ MPa m$^{-\frac{4}{3}}$ a$^{\frac{1}{3}}$ | 50 m |

**Table 2.** Model resolutions used in the current study.

| Resolution | Number of elements in the horizontal | Element size in the horizontal |
|---|---|---|
| R0 | 250 | 7.2 km |
| R1 | 500 | 3.6 km |
| R2 | 1000 | 1.8 km |

**Table 3.** Summary of experiments. The experiment name is given in bold for experiments whose results are analysed in the current study. Experiments not given in bold provide spinup/initialisation for those analysed. The basal melt forcing is described in Section 2.2 and the experimental design in Section 2.3. $W$ is parameterised channel width, $S_w$ is the water column thickness scaling of basal melt (equation 8), and $\Omega$ is a basal melt tuning parameter.

| Experiment | Description | Initial condition | $W$ | $S_w$ used? | $\Omega$ |
|---|---|---|---|---|---|
| SPIN | initial SPIN up | Uniform slab ($H = 300$m) | 1000 km | - | - |
| ADVA | ADVAnce due to buttressing increase | SPIN final state | 150 km | - | - |
| **RB** | Retreat due to Buttressing reduction | ADVA final state | 1000 km | - | - |
| **RHW** | Retreat due to High melt (with Water column scaling) | ADVA final state | 150 km | Yes | 0.045 |
| RHWB | Retreat due to High melt (with Water column scaling) and Buttressing reduction | ADVA final state | 1000 km | Yes | 0.045 |
| **RLW** | Retreat due to Low melt (with Water column scaling) | ADVA final state | 150 km | Yes | 0.009 |
| **ALW** | Advance due to Lowering of Melt (with Water column scaling) | RHWB final state | 150 km | Yes | 0.009 |
| RHB | Retreat due to High melt (no water column scaling) and Buttressing reduction | ADVA final state | 1000 km | No | 0.045 |
| **RL** | Retreat due to Low melt (no water column scaling) | ADVA final state | 150 km | No | 0.009 |
| **AL** | Advance due to Lowering of melt (no water column scaling) | RHB final state | 150 km | No | 0.009 |

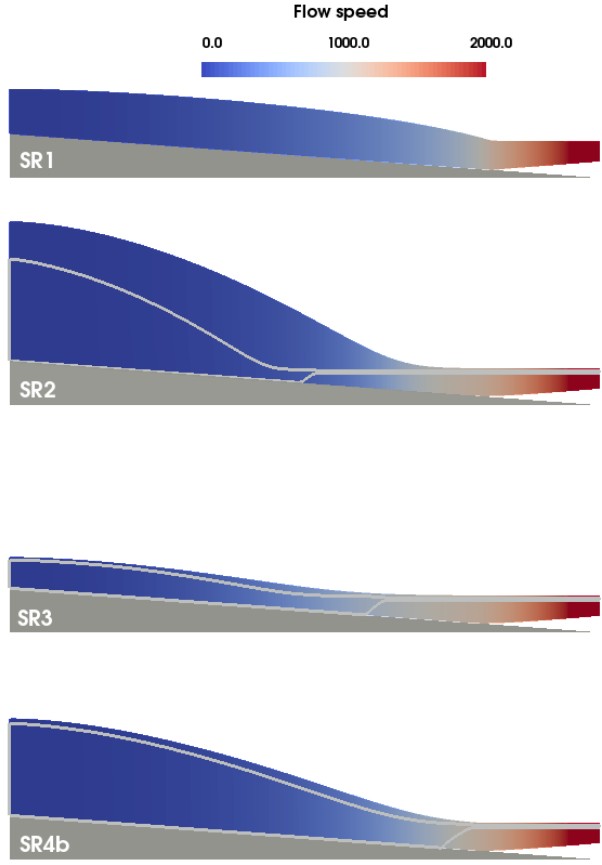

**Figure 1.** Ice geometry and velocity magnitude (m a$^{-1}$) at steady state from the ADVA experiment. Resolution R1 is shown, but these profiles do not vary significantly with resolution. These profiles provide the starting point for the retreat simulations. The final state of the melt-induced experiment RHW is overlain in grey outline. Bedrock is shaded in grey. Vertical exaggeration is 150 times.

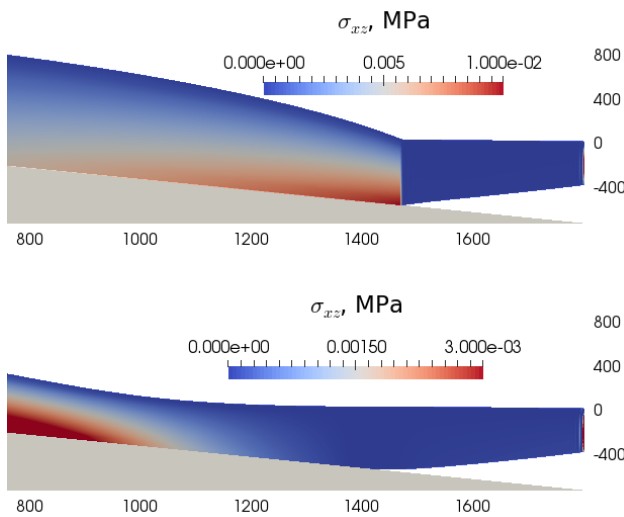

**Figure 2.** Stress tensor component $\sigma_{xz}$ shown at the end of the ADVA experiment for SR1 (top) and SR3 (bottom) for the seaward 1000 km of the domain. At the base of the ice this approximates the basal shear stress, given the low slope of the bed. Note the different colour scales. Distance from the ice divide is shown along the bottom in km. Height relative to sea level is shown at the right end of the plots in m. Bedrock is shaded in grey. Vertical exaggeration is 200 times.

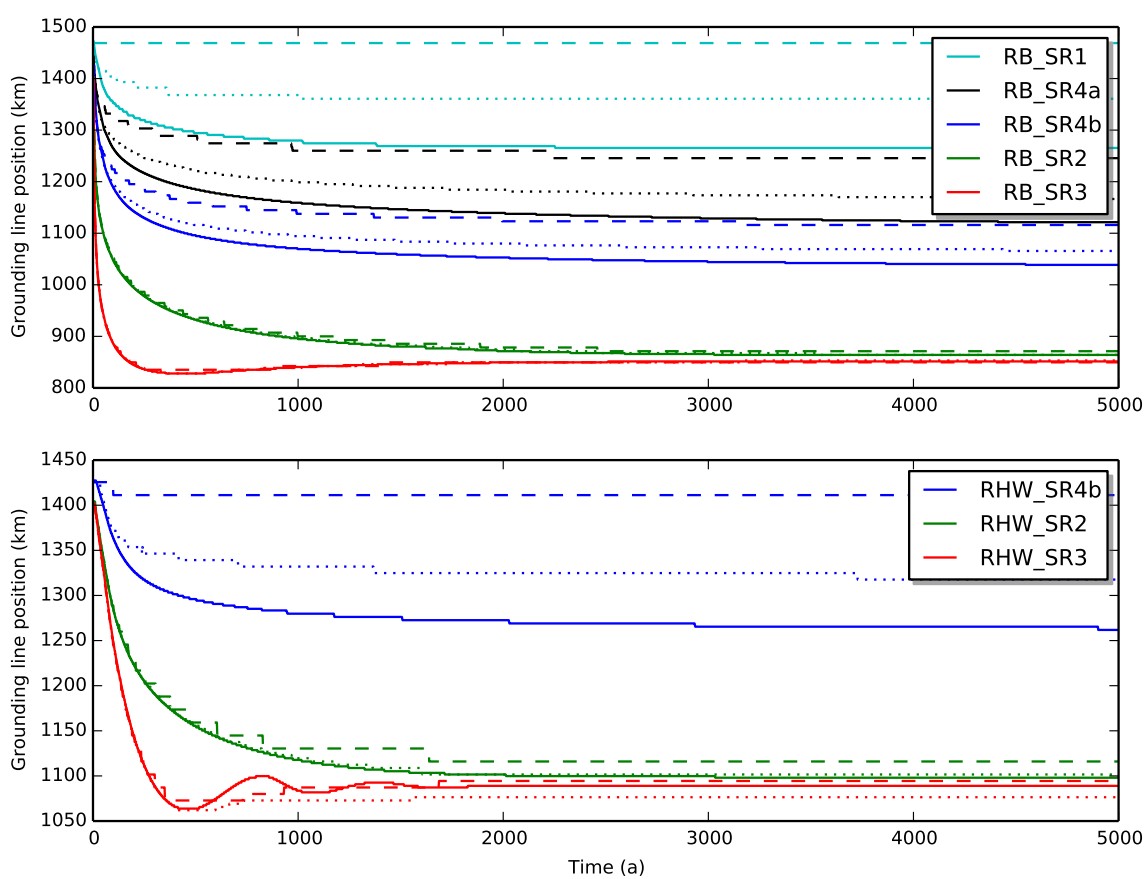

**Figure 3.** Evolution of grounding line position relative to the inland boundary during retreat simulations with different sliding relations. Sliding relations are described in Section 2.1 and Table 1. Experiments are described in Section 2.3 and Table 3. Resolutions (Table 2) are coarse (R0, dashed line), medium (R1, dotted line) and fine (R2, solid line). The vertical ordering of the families of curves matches that of the legend tables.

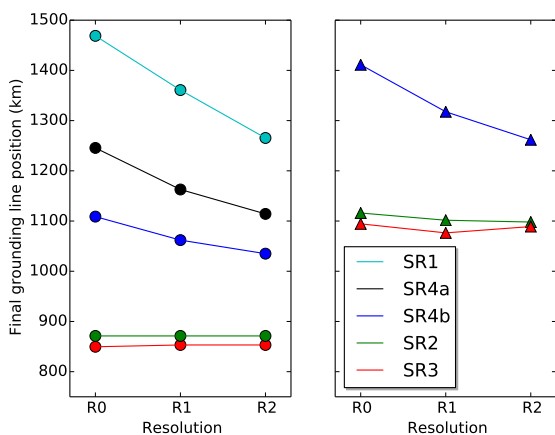

**Figure 4.** Final grounding line position (after 40 ka) against resolution (Table 2) for the retreat due to buttressing reduction (RB, left panel) and retreat due to basal melting (RHW, right panel) with the different sliding relations (Table 1). The y-axis range is identical in both panels. The vertical ordering of the families of curves matches that of the legend tables.

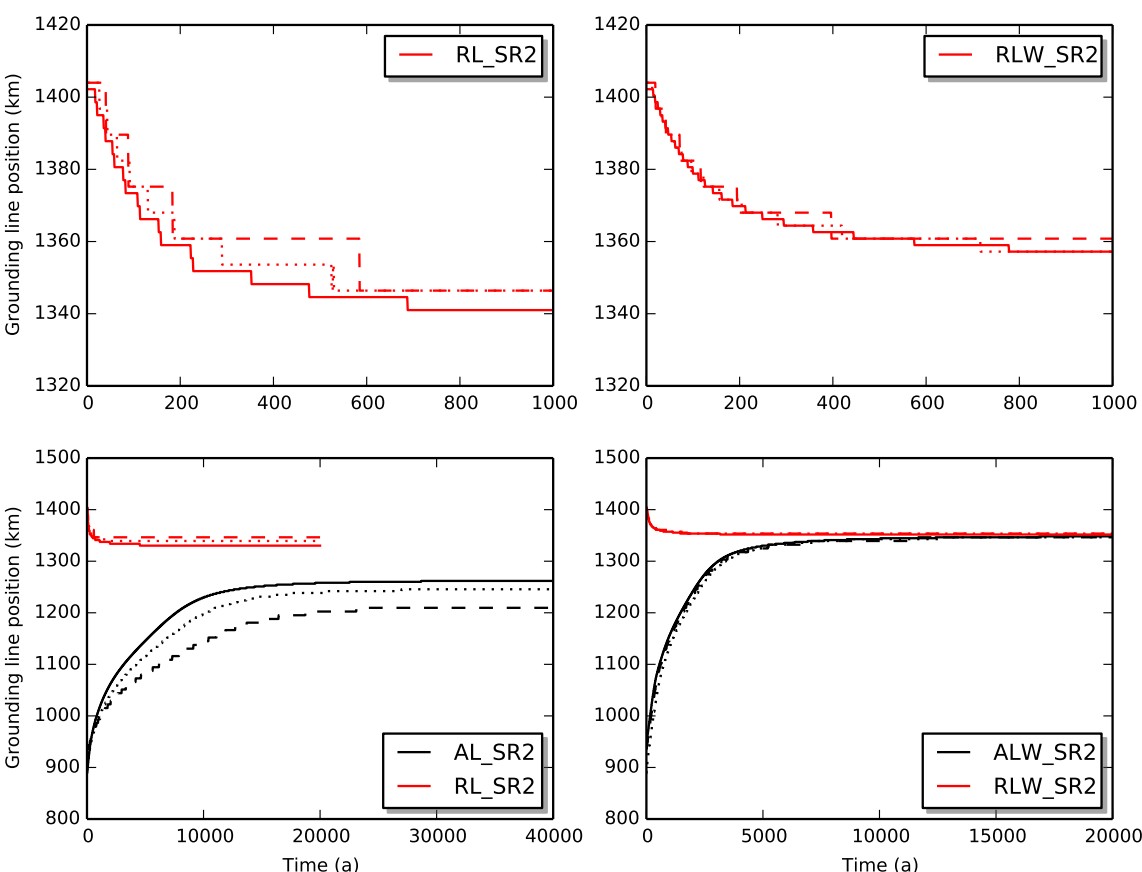

**Figure 5.** Evolution of grounding line position during the sub-shelf melting simulations with effective pressure dependency in the basal sliding relation. Right hand panels show results with the water column scaling factor $S_w$, active. Left hand panels show the abrupt melting transition. The upper panels show the detail of the early stages of the retreat simulations, whereas the lower panels show the full simulations. The sliding relation (SR2) is described in Section 2.1 and Table 1. Experiments are described in Section 2.3 and Table 3. Resolutions (Table 2) are coarse (R0, dashed line), medium (R1, dotted line) and fine (R2, solid line).