# Peer review of "Marine ice sheet model performance depends on basal sliding physics and sub-shelf melting"

_The Cryosphere, 2016_

## Referee Comment (RC1) · VC Tsai (Referee) · 8 Aug 2016

In their manuscript, the authors describe a couple of numerical tests that show how the resolution required of the numerical ice sheet model Elmer/Ice depends on the way in which basal sliding and sub-ice shelf melt rates are parameterized. The results shown seem to agree with previous suggestions that the required resolution should depend on how smoothly the stresses change across the grounding zone, and the current manuscript demonstrates this, making it a useful contribution. However, I had a number of significant concerns about both the numerical results themselves as well as the general applicability of the results to ice sheet modeling. On the numerical side, it seems impossible for the steady-state spin up simulations to achieve a resolution-independent

grounding line whereas the steady-state after forcing is resolution-dependent since both are just steady-state solutions. This strange result brings into question the validity of the results and needs to be explained. On the applicability side, since the parameterizations used are not physically motivated, it is not clear if the results are truly relevant for real ice sheets. The authors should at least discuss this point. In addition to these two major points, the main results could be better illustrated, especially Fig.3 in which more detailed information could easily be provided. Once these points are addressed, the manuscript could be a nice contribution to the literature. More specific comments are below.

The one major concern I have regarding the numerical results is how the steady-state spin up simulations can be resolution independent whereas the forcing steady-states are resolution dependent. My understanding is that the spin up simulations should be just one particular choice of a forcing experiment, where the steady-state solution is achieved by the end of the spin up. If this is true, the same 'forcing' that is used to create the spin up should result in the same steady-state solution as when used in a forcing experiment. Thus, at a given resolution, the solutions should be the same, whether they agree or disagree with solutions at other resolutions. Apparently, this understanding is incorrect since the spin up simulations are claimed to generally have the same steady state but the forcing steady states do not. Since this defies expectation, the authors need to explain this, i.e. why the spin up simulations cannot be thought of as a particular forcing (or in other words, why is the end of a retreat/advance simulation not equivalent to the end of a spin up simulation?). The authors do state on p7L15 that the melting forcing may cause the resolution dependence, but this does not explain Fig3a. It may also be useful if the authors are more precise in their statement on p6L6 that the spin up simulations "do not vary significantly with resolution". How significantly? Are they within a single grid spacing?

The other significant concern I have is regarding how applicable the results are. The authors choose to test 3 sliding laws (SR2-4) that were proposed 30 years ago, where

the effective pressure dependence is inserted in an ad hoc manner, despite the existence of at least 2 more recent parameterizations of basal sliding (Schoof 2005 and Tsai et al. 2015) that are more physically motivated. Perhaps it is easier to test the 3 sliding laws used, but if they do not describe the physics properly then they are not relevant to ice sheets. At a minimum, the authors should at least explain why they choose the 3 sliding laws and comment on whether the results can be expected to hold more generally or not. For the melt parameterization, the authors also seem to arbitrarily choose a smoothly decreasing melt rate for no apparent reason. In reality, if anything, at many grounding lines, the melt rate is expected to increase rather than decrease, so the relevance of a smoothly decreasing melt rate is not clear. Are there ice shelves where basal melting decreases towards the grounding line? This should be commented on.

Additional comments:

P2L17: Feldmann et al. 2014 should be cited in this paragraph as well.

P3L1: "The starting point" could be rephrased for clarity.

P3L9: Missing half sentence.

P3L16: Since the C's in the different equations are actually different, different letters (subscripts?) should be used. Otherwise it is confusing.

P5L8: Sentence wording is confusing. Sounds like a given model's horizontal resolution is spatially variable, rather than what is intended with 3 different models with different resolution.

P5L21: Should this be 1000km rather than 150km?

P6L12: There are multiple shear stresses. Need to specify which one.

P6L15: Throughout this section, it would be useful if the authors made it easier for the reader to follow which simulation is referred to. For example, the acronyms could

be spelled out once in the main text, the main text could refer to the colors in the figures, and also refer specifically to the figures and figure panels in which the results are shown.

P6L15: Whether or not the simulations (particularly SR2 and SR3) actually achieve resolution independence is difficult to see in Figure 3, partly because it is hard to see the step size. Since the steady-state solutions are the only ones that need to be compared, and contain the useful quantitative information, I would suggest making subplots for each SR# experiment that plot zoomed-in steady-state grounding line location (on y-axis) vs. mesh resolution (on x-axis). It would be preferable if there were more than 3 points, but if that is all that is computationally feasible, that is understandable.

P8L12: Sentence is confusing. In fact, much of this page (L3-L20) is not coherent, and I would suggest the authors reword for clarity and flow.

P8L29: As commented earlier, since the sliding laws used are not physical, it is not clear that the results are easier to interpret than those of the Gagliardini study.

Fig.1: The point of Fig.1 is not entirely clear, and the gray overlay makes it impossible to see what is stated about there being little vertical shear.

---

## Referee Comment (RC2) · J. Bassis (Referee) · 11 Aug 2016

Overview:

This paper shows that grounding line migration can be simulated with coarse resolution marine ice sheet models that implement sliding laws that force the basal shear strength to smoothly decrease to zero and force basal melt to smoothly decrease to zero near the grounding line. This study builds upon–and sometimes challenges–the growing body of literature that examines numerical accuracy and convergence in ice sheet models. The premise of this study is that the grounding line position (and migration) determined in numerical marine ice sheet models can depend greatly on model resolution. This raises the concern that grounding line position ultimately depends (unphysically)

on grid resolution. This is an important topic in glaciology and ice sheet modeling that is appropriate for The Cryosphere. Overall, despite some lengthy questions about aspects of the study, suggestions for some additional work and stylistic/figure improvement, I think this study is an important contribution to this growing field. Below I outline some points that I found confusing along with some suggestions for improvement.

Concerns

1. What is meant by "grid scale dependence"?

Models usually display "grid scale dependence" for one of two reasons. The first reason is that the resolution of the model fails to appropriately resolve the fundamental scale in the physics. This leads to a numerical model that fails to accurately represent the underlying physics and, not surprisingly, inaccurate numerical solutions. From a physicists perspective, there is little of physical interest in this situation and the poor accuracy can be cured by merely increasing the model resolution until the fundamental scale is appropriately resolved. From the perspective of a numerical modeler however, resolution may be severely limited by computational resources and increasing the numerical resolution sufficiently may be challenging, require clever tricks and/or much bigger computers. Sadly, most climate models live in this regime and this often necessitates clever parameterizations for sub-grid scale processes. A second reason a model may display "grid scale dependence" occurs when the model results actually fail to converge (or even diverge) as resolution increases. This leads to a true grid scale dependence where no amount of grid refinement can cure the divergence. This is, of course, unphysical, and usually caused by a breakdown in the model physics. This can point to a problem in the equations solved (e.g., ill posedness) or can result from a model that has been applied to a length scale where the model no longer applies. For example, the stress field near sharp cracks in elastic mechanics formally diverges. This is unphysical and actually leads to poor numerical convergence in numerical models that don't explicitly account for the crack tip singularity. The singularity in the stress field is caused by a break down in the elastic assumption and can be cured by modifying the physics to include a plastic process zone near the crack tip. This example is relevant because in the elastic crack case numerical models simulations will exhibit grid-scale dependencies. This dependency can be cured by either incorporating the singularity explicitly in the numerical scheme (appropriate for length scales larger than the process zone) or by physically curing the singularity by modeling the process zone (appropriate for length scales on the order of the process zone or smaller).

The reason for this long digression in my review is that it is unclear to me which type of grid scale dependence the authors are invoking. Previous work (see, e.g., the work by Schoof on grounding zones) suggests that the problem with resolving grounding zones is "merely" a numerical resolution issue and not a physical issue. Some of the statements by the authors instead seem to indicate that they think the sliding law and basal melt parameterizations themselves are to blame. I think readers would appreciate a more precise discussion of the type of grid-scale dependence and its connection to physics. My understanding is that the authors are primarily discussing the first type of grid-scale dependence, but the relationship between the grid-scale dependence and incorrect model physics would be improved.

2. Physical appropriateness of the sliding laws: The previous point leads me to a more fundamental question: the authors introduce 4 sliding laws. The first is the standard "Weertman" sliding law often used in ice sheet models. The remaining sliding laws represent various heuristic generalizations of Weertman sliding to include effective pressure. I have no objection to picking a subset of the infinitely large number of possible sliding laws. However, what is missing is a discussion of why the authors think the various generalizations of Weertman sliding are more appropriate for more realistic ice sheet models. This is especially true when considering the simple assumption employed by the authors that effective pressure is a linear function of water depth **all the way to the ice divide**. There are adequate measurements at this point to show that effective pressure is not simply related to bed topography. As the authors clearly state, subglacial hydrology is beyond the scope of this study. Nonetheless, it is unclear to

me why a sliding law that incorporates an unrealistic effective pressure is always more accurate or appropriate than a model that ignores effective pressure. In other words, why is it better to assume effective pressure varies like water depth as opposed to the (implicit) assumption in the Weertman sliding law that effective pressure is constant? To be clear, effective pressure proportional to water depth is often invoked because it forces basal shear stress to decay to zero at the grounding line, but there is no clear physical reason why this decay is required to occur over a length scale determined by bed topography as opposed to basal hydrology (or other processes). The fact that one sliding law is more convenient for numerical modelers doesn't necessarily make it the most physically appropriate sliding law. See point 1. Here I think the authors case would be strengthened considerably if they could provide first principles or–I think this is easier–empirical reasoning to suggest one or more of the sliding laws is more plausible than others. For instance, the 4 sliding laws predict very different ice sheet profiles with different divide thickness, slopes and even concavity of the profile. Can we say anything about the reasonableness of the sliding law assumptions based on observed ice sheet profiles (Pine Island?, Thwaites?, Lambert Basin, Siple Coast Ice Streams?). My guess is that one might find different sliding laws appear more appropriate for different portions of the ice sheet? I understand the experiments are necessarily idealized, but the argument that the results presented here are relevant to actual marine ice sheets will be much stronger if the the authors could show that the parameter regime they examine resembles an actual ice sheet.

3. Selection of numerical parameter: Another question along a similar line of reasoning is how did the authors select the numerical "C" coefficients in the sliding laws? Are they chosen so that the average basal shear stress across the ice sheet is the same for all models? So that the basal shear at a particular velocity and effective stress matches? So that the profiles are as similar as possible? So that the ice sheet profiles are qualitatively similar to some section of an ice sheet? I can imagine that it may be possible to take a single sliding law, like the Weertman relation (SR1) and obtain different convergence results depending on the value of the sliding coefficient. (The

boundary layer size is a function of the sliding coefficient and so one could presumably make the problem easier or harder based solely on varying the C value.) Are the results that the authors show a function of the form of the sliding law or the magnitude of basal friction (or both)?

4. More detailed explanation of the results: Getting into the actual experiments, I'm puzzled by the fact that the authors spin up the models and find a steady-state that is independent of resolution, but then find that some of the forcing experiments do yield a dependence on grid resolution. Is this because the models only display grid resolution dependence when buttressing or basal melt is implemented as a forcing and not otherwise? Either way this would appear to indicate that grid sensitivity depends not only on the parameterizations considered (basal melt and sliding), but also on the type of forcing. This suggests that dependence on grid resolution is far more complicated than indicated and depends not only on the model physics, but also on the external factors driving change. I would like to see this commented on and explained in more detail in the manuscript.

5. How generalizable are the results: Similarly, I would urge the authors to dig a bit deeper into the physics behind the results. The authors perform a suite of idealized experiments using a particular geometry, set of parameterizations (sliding and basal melt) and forcing and find some of these combinations are more amenable to simulation in low resolution ice sheet models than others. It is, however, not straightforward to translate these results to other, more physically realistic situations. Would the results hold if instead of water depth dependent effective pressure, a model calculated effective pressure based on a subglacial hydrology model? Here I strongly encourage the authors to dig deeper into their results and guide us in interpreting them. I suspect that many of the results can be anticipated by a analytic consideration of the structure of the transition zone near the grounding line for the different sliding laws. My (perhaps naive) expectation is that models need a grid spacing that is small compared to the length scale of the transition zone between vertical shear and horizontal spreading

dominated flow regimes.

Comments on presentation

The writing in the manuscript is generally clear, but stylistically, the manuscript contains many paragraphs that consist of one or two sentences. The abstract, for example, contains 4 distinct paragraphs! Normally, abstracts are a single paragraph. Traditionally, at least in the US, paragraphs are units that contain coherent ideas. Paragraphs depend on each other, but paragraphs themselves should be coherent. I found it jarring to have indentation that has little relation to the organization of ideas. For example, on Page 4 line 9 contains a paragraph with the single sentence starting with "This parameterisation is similar to that used in the . . ." The next paragraph begins with a definition of a variable. Neither of these are coherent or can stand on their own. This peculiar isolation of sentences into incoherent paragraphs was jarring to my American sensibilities, but this is perhaps a European or Cryosphere Discussion stylistic preference. I leave it at the editors and/or authors discretion as to whether this is stylistically appropriate.

I was also flummoxed by the bestiary of acronyms. I was confronted with ISM, MIP, MIS all in the first several paragraphs and later hard to deconstruct SR1 and SR2, ALMW, ALMN amongst what seemed like many others. I have strong personal preferences to avoid all unnecessary acronyms and, unless the authors are forced to pay by the word, I urge the authors to use complete words instead of acronyms when ever possible. If this is inconvenient or impossible, then I would suggest a table that readers can easily refer to that provides us with a dictionary of acronyms that we can conveniently look up.

My last comments on presentation concerns the figures, which although physically appealing, were difficult for me to parse. For example, I struggled to understand Figure 1. According to the caption, panels show the steady and final states for the basal melting experiment. However, the top panel doesn't have the gray shaded profile. Why not? Moreover, given that the shaded gray profiles obscure parts of the initial profile, I

would prefer to see this figure broken into two figures. The first would show the initial, post spin-up profiles. The second would show the final, post simulation profiles (with velocities). This would allow readers to more easily be able to see the differences in the shapes of the initial and final profiles and contemplate the role that the sliding law in modifying the morphology of the profiles and the transition zone. This figure also needs a scale bar to show horizontal distance (and possibly vertical elevation).

For Figure 2, why are only 2 steady states shown in Figure 2? Why not show all of them (as in Figure 1)?

Figure 3 was difficult for me to parse. I would urge the authors to considering using color schemes that are more appropriate for the color blind and/or line types that don't depend as much on color.

Typographical comments:

page 1 near line 10: even with –> even with

page 3 above line 10: incomplete sentence "These values for p and q are chosen for simplicity, and deviate" ????

From a physical perspective, Equation 2 is problematic. It assumes that the ice sheet has a basal hydrology sentence that is perfectly connected with the ocean all the way to the ice divide.

The constant "C" in Equations (3)-(6) has different units in the suite of basal drag parameterizations. To avoid confusion, I recommend using a different symbol or appending subscripts to more clearly indicate that the numerical value of C (and its units) are not identical in all experiments.

Line 4, line 15 and elsewhere: need a space between the number and unit "100m" –> "100 m"

After section 2.3 "1800km" –> "1800 km"

Accumulation rate: What is the motivation for the accumulation rate defined by Equation (11). This appears to predict linearly increasing accumulation with zero accumulation at the ice divide? Why not use a constant accumulation, which would seem to be more realistic for much of Antarctica?

Spin-up is independent of resolution. This seems to indicate that resolution only matters in some experiment types???

Page 6, line 6 "focusses" –> "focuses"

Page 6, line 7 missing commas –> "The spinup simulations do however vary" –>The spinup simulations do, however, vary"

Page 6, line 21: "But even SR4b still shows significant resolution dependency." –> sentence fragment, consider revising.

Page line 23: "Since" indicates time (e.g., I haven't slept since yesterday). I think the authors want "Because" (Because SR1 and SR4a . . .)

Page 7, line 4 missing space: "20ka" –> "20 ka"

Page 7, line 29, "Any resolution dependence in a model is inevitably non-physical. Ideally model behavior should converge with finer resolution." This line is perplexing and indicates my fundamental misconception. Are the authors arguing that grounding zone position fails to converge with increasing resolution and is, hence, resolution dependent or are they arguing that grounding line position requires finer resolution than they have available. There is no particular guarantee that a numerical model with a priori specified resolution will be an accurate representation of a given set of partial differential equations. This is especially the case when the resolution of the model is more coarse than the fundamental scale of the system.

Page 8, Line 10-15: I don't think it is true that basal melting *requires* subglacial discharge. Because the ice near the grounding line is (usually) located at a depth that is much greater than the melting point of sea water, one expects melt near the

grounding line of ice shelves with deep grounding lines, even in cold cavity ice shelves. In the traditional ice-pump theory, the cold melt water mixes with sea water and forms a buoyant plume. I'm puzzled by the argument that subglacial discharge is required to initiate the process? Are the authors arguing for massive super cooling near grounding lines? Is this usually observed? Models of submarine melt under ice shelves rarely include grounding discharge and yet predict realistic patterns of basal melt. Why is this if submarine discharge is a crucial component of the process? It also seems like the relevant length scale over which basal melt must go to zero is going to related to the characteristic width of the buoyant plume. Can this be estimated and used to better constrain the parameterization of basal melt?

---

## Author Comment (AC1) · 11 Nov 2016

**Author's response to reviewer V Tsai.**

An abbreviated version of the reviewer's original comments is given in standard font, and our reply in italics.  The full review is linked here:
http://editor.copernicus.org/index.php/tc-2016-149-
RC1.pdf?_mdl=msover_md&_jrl=25&_lcm=oc108lcm109w&_acm=get_comm_file&_ms=52
435&c=110422&salt=2038061631095973250

*Authors general comments.  The review from Dr Tsai was useful and we'd like to thank him. In the cases where we do not fully agree with Dr Tsai's criticism, his review enabled us to modify the text to make clearer our arguments.  In particular some of our justification for our methods was embedded in the discussion in a way that wasn't very obvious to the reader, and we have moved some of this to the methods sections and hopefully clarified our arguments.*

**Dr Tsai's general comments**

Dr Tsai had two main concerns – one about the numerics, and another about the applicability of the results and the relevance of the sliding relations considered.
A third issue concerned the melt parameterization.

**Numerics:**
"On the numerical side, it seems impossible for the steady-state spin up simulations to achieve a resolution-independent grounding line whereas the steady-state after forcing is resolution-dependent since both are just steady-state solutions. This strange result brings into question the validity of the results and needs to be explained."

*Dr Tsai is correct in his recognition that the uniqueness of steady state solutions does not hold here.  This issue is not unique to this study.  While the corresponding real (continuum) system (if there were a real world glacier on a linear down-sloping bed) would almost certainly have only one viable steady state, ice sheet models, especially at coarse resolution, typically feature a region of locally stable steady states for a given forcing, and this region shrinks as resolution is refined.  There is a discussion of a region of locally stable grounding line positions in Gladstone 2010 (JGR).  That 2010 paper uses a flowline SSA model, but this issue of advance and retreat experiments not matching has been seen also in other models, and we are not referring hysteresis in the presence of an overdeepened bed.  For example see Fig 3 in the MISMIP paper (Pattyn 2012).*

*We've added a couple of lines about this where spinup is mentioned in the results section.*

**Applicability:**
"The authors choose to test 3 sliding laws (SR2-4) that were proposed 30 years ago, where the effective pressure dependence is inserted in an ad hoc manner, despite the existence of at least 2 more recent parameterizations of basal sliding (Schoof 2005 and Tsai et al. 2015) that are more physically motivated."

To the best of our knowledge none of the sliding relations in our paper was actually in use 30 years ago, although they are certainly motivated by developments back then, which recognised the deficiencies of the simple Weertman sliding relation in describing fast sliding flow, particularly in marine ice sheets.

Several researchers explored the introduction of an effective pressure denominator into sliding relations (e.g. BIndschader 1983). In the context of the fast flowing ice streams and outlet glaciers of the marine West Antarctic ice sheet, this was a response to the character of those flows: increasing velocity towards the grounding line despite steadily decreasing surface slope and driving stress. The linear relation between basal shear stress and effective pressure of SR2  was suggested by McInnes and Budd 1984, specifically to treat sliding in the approach to the grounding lines in West Antarctica, again as an effort to reconcile velocities, driving stresses and the height above floatation (although their velocity exponent in equation (1) was linear!). SR3 and SR4 are modifications suggested in 2014 for  this paper. The motivation of SR3 is to provide the same behaviour as SR2 near the grounding line but to interpolate to the standard Weertman type relation further from the grounding line, where the concept of connecting basal water pressure to the ocean conditions would seem less justified.

This suggestion of historical irrelevance is beside the point – publication year is not a strong indicator of scientific justification.  Glen's flow law is older than these sliding laws, was empirically derived, and is widely used today although more complex alternatives are available.

Having said that, we turn to responding to the claim that more recent sliding laws, stemming from the work of Christian Schoof in 2005, are "more physically motivated".  The Schoof (2005) work is indeed physically motivated.  It is a theoretical derivation based on certain assumptions.  Those assumptions include the bedrock being of hard rock rather than deformable sediment and considering "cavitation" (another "old" idea dating back to Lliboutry) where the ice base locally detaches from the bed in the lee of obstacles.  Even if the work of Schoof and the assumptions it contains are perfect from there on in, these initial assumptions certainly do not hold for the whole of Antarctica.  For example, both Schoof 2005 and Gagliardini 2007 (following Iken 1981) make the assumption that tangential stresses between the bed and the ice are zero, assuming that there is always a thin film of water present, and thus the stress parameterisation arises entirely from normal stresses (what is called form drag in other flow situations) - which is why the max slope of the cavities is so important in determining Iken's bound.  It is not clear to us that these assumptions will always hold true.

This is perhaps an appropriate point to comment that the criticism in Schoof (2005) regarding unbounded flow relations (of the form of our equation 1) is not about the presence of the effective pressure per se – it is about the risk of unrealistically high shear stresses arising through power law relations. Indeed that paper is not concerned directly with the main issue at the grounding line – the vanishing of effective pressure. Furthermore, it is clear that any relation with positive powers of effective pressure in equation (1) has a far better chance of satisfying a Iken (1981) or Schoof (2005) boundedness requirement on $\tau_b/N$ for finite values.

Regarding the interesting parameterisation in Tsai et al 2015 - this certainly provides a "Coulomb" limiting cutoff, but apparently simply assumes that the Weertman sliding relation

*is otherwise completely correct. Several researchers, such as the early researchers on Antarctic ice stream flow such as Bindschadler (1983), Budd and his group, and others, and more recently Schoof (e.g. Schoof (2011) Journal of Fluid Mechanics) seem less convinced that the general character of basal sliding is settled.*

*The form of the effective pressure dependence in the Budd laws was motivated from the output of laboratory experiments of ice sliding (Budd et al 1979). This starting point was taken forward in various different parametrisations in that group's pioneering Antarctic modelling – informed by what the then available data showed about connections between velocities, surface slopes, ice thickness and bedrock geometry. We think it is much fairer to say "empirically derived" rather than "Ad Hoc" in this case (was Glen's law not also empirically derived based on laboratory experiments?).*

*So on the one hand we have an empirically derived law and on the other we have a theoretically derived law. It seems clear to us that arguments could be made both ways. It also seems likely that the cavitation sliding law is not always going to hold true, and although Schoof's contribution to sliding relations is excellent work, we don't think it benefits the community as a whole for everyone to always use the cavitation law. At least not until there is a lot more direct evidence from real world applications to support it.*

*Having said all that, the choice is to some extent arbitrary, since the differences between these laws are smaller than the differences between parameter choices for a given law, i.e. I can choose parameters such that both laws look pretty similar, or I can make two different parameter choices within one law that give massive differences. Also the two laws are closer to each other than they are to Weertman sliding, and the choice of SR2-4 here give us control over the way basal drag is smoothed across the grounding line.*
*Given the weak dependence on sliding velocities in our various sliding relations, their local behaviour at the grounding line may not be that dissimilar to a Coulomb relation there.*

*In summary, it is not at all clear to us that any one of these sliding laws is "right" and another "wrong", and there are other sliding laws that we haven't considered at all here, but we don't want to see everyone in the community using the same law until we're really sure that it is always "right". We think that this discussion is far too long to put into the paper. We modified the paper to briefly discuss other sliding relations, to try and give perspective to refute the suggestion that 30 years ago people just made things up "ad hoc" and also to make the point that we are not advocating any particular sliding relation as being "better" than others just because we are using it, in the basal sliding methods section.*

**Basal melt parameterization:**
"For the melt parameterization, the authors also seem to arbitrarily choose a smoothly decreasing melt rate for no apparent reason."

*We consider it is quite clear why we present simulations both with and without the smooth decrease in melting near the grounding line. A smooth decrease is an opposite member to abrupt change, and appropriate for what is essentially a sensitivity study. There is of course a physical justification for high variation in amount of melting near the grounding line: varying strength of sub-glacial outflow, and this is already mentioned in the discussion.*

*However, it seems from Dr Tsai's comment that it would be appropriate to put some justification in the methods section where the basal melting is described, so we have done this.*

*We hope it was clear that the melting parametrisation gives greater melting at deeper ice shelf drafts via the "thermal driving" , to which we added a physically motivated optional factor that might reflect reduced heat transport in a sub-ice shelf cavity of limited water column thickness.*

*We also changed the MISOMIP reference from the GMD discussion paper to the peer reviewed one, since this is now published.*

**Response to line by line comments.**

P2L17: Feldmann et al. 2014 should be cited in this paragraph as well.

*Why?  The context here is effective pressure dependency in sliding relations.  I like the Feldman 2014 paper, but it is not about pressure dependency (or were there 2 Feldman 2014 papers?).*

P3L1: "The starting point" could be rephrased for clarity.

*We changed the wording here.*

P3L9: Missing half sentence.

*Oops!  Thanks, we finished the sentence.*

P3L16: Since the C's in the different equations are actually different, different letters (subscripts?) should be used. Otherwise it is confusing.

*We've made this change, but have minor reservations because it doesn't really help with Table 1.*

P5L8: Sentence wording is confusing. Sounds like a given model's horizontal resolution is spatially variable, rather than what is intended with 3 different models with different resolution.

*Dr Tsai is right, thanks. Wording improved.*

P5L21: Should this be 1000km rather than 150km?

*Good point, fixed.*

P6L12: There are multiple shear stresses. Need to specify which one.

*This is sigma_xz, and this should be clearer now after minor changes to the text.*

P6L15: Throughout this section, it would be useful if the authors made it easier for the reader to follow which simulation is referred to. For example, the acronyms could be spelled out once in the main text, the main text could refer to the colors in the figures, and also refer specifically to the figures and figure panels in which the results are shown.

*We've changed the experiment acronyms to make them easier to remember and more intuitive. We've also introduced most of them in the text (experiment design section). We also refer to Fig 2 line colours in the text (results section).*

P6L15: Whether or not the simulations (particularly SR2 and SR3) actually achieve resolution independence is difficult to see in Figure 3, partly because it is hard to see the step size. Since the steady-state solutions are the only ones that need to be compared, and contain the useful quantitative information, I would suggest making subplots for each SR# experiment that plot zoomed-in steady-state grounding line location (on y-axis) vs. mesh resolution (on x-axis). It would be preferable if there were more than 3 points, but if that is all that is computationally feasible, that is understandable.

*A new figure has been created as suggested.*

P8L12: Sentence is confusing. In fact, much of this page (L3-L20) is not coherent, and I would suggest the authors reword for clarity and flow.

*Re-reading this it seemed fairly clear. We've made some modifications to this section, including removing some bits that were not essential to the argument we are trying to make. Hopefully this is now clearer.*

P8L29: As commented earlier, since the sliding laws used are not physical, it is not clear that the results are easier to interpret than those of the Gagliardini study.

*We don't understand this comment. We don't see where in that paragraph we claimed that our results are easier to interpret.*

Fig.1: The point of Fig.1 is not entirely clear, and the gray overlay makes it impossible to see what is stated about there being little vertical shear.

*Figure 1 seems to generate mixed opinions. It aims to show the different profile shapes due to different sliding relations. We have changed the grey to only outlines, and added the bedrock.*

---

## Author Comment (AC2) · 11 Nov 2016

**Author's response to reviewer J Bassis.**

An abbreviated version of the reviewer's original comments is given in standard font, and our reply in italics. The full review is linked here:
http://editor.copernicus.org/index.php/tc-2016-149-RC2.pdf?_mdl=msover_md&_jrl=25&_lcm=oc108lcm109w&_acm=get_comm_file&_ms=52435&c=110576&salt=1162785007638623258

*Authors general comments. The long review from Dr Bassis gave plenty of food for thought and we'd like to thank him. There were a few places where Dr Bassis clearly thought that we were arguing a stronger point than we had intended (sometimes not fully defensibly). This helped us modify the text to avoid our remarks being interpreted as implying stronger statements than we intend. Some of his other suggestions could be fruitfully pursued as research projects, but seem to range rather far from the focus of our current paper.*

**Dr Bassis' general comments**

There were five major points of concern:

**1.**
Models usually display "grid scale dependence" for one of two reasons. The first reason is that the resolution of the model fails to appropriately resolve the fundamental scale in the physics. A second reason a model may display "grid scale dependence" occurs when the model results actually fail to converge (or even diverge) as resolution increases.

*We are referring to the first kind of grid scale dependence referred to by Dr Bassis, in which the difficulties of resolving the physical problem cause strong resolution dependence, and this resolution dependence is expected to converge with resolution. We don't see how discussing the other kind of resolution dependence (in which convergence does not occur) can strengthen the paper. Clearly we needed to make the aims of our paper clearer, and we hope that revisions have achieved that, without specifically discussing the second type of resolution dependence. Our goal is to explore the degree of resolution refinement necessary to get consistent behaviour in our simulations – effectively convergence to required precision – for a variety of sliding relations (boundary conditions for basal traction under grounded ice), and for gradual or abrupt imposition of ice shelf basal melt rates. Both these constitute parameterisations - of unresolved small scale basal processes in ice sliding, and of the delivery and transfer of heat from the sub ice shelf ocean in the case of basal melting. Some of our variants of these parametrisations proved amenable to achieving sensible modelling outcomes at much coarser resolutions than usually considered necessary in such studies. In other cases we demonstrate that much finer resolution is required. Unsurprisingly the need for much finer resolution generally emerges when there are discontinuities in these two basal boundary conditions.*

*The lead author has worked with several ice sheet models, all of which do appear to converge, and the reason why some are worse than others at a given resolution is due to the physical problem itself, and the approaches taken to deal with the problem. Essentially, a*

smooth change in a forcing field (be it basal drag or sub shelf melting) can be much better approximated at coarse resolution than an abrupt change.

*The lead author has some familiarity with Schoof's work, and the convergence problems described can indeed be viewed as a purely numerical issue. That does not mean that it is not also a physical issue. With a different choice of physics that same numerical issue becomes more tractable at a coarse resolution, which is one of the main points of this paper. Neither our work nor Schoof's work is advocating a particular sliding law as being "correct". Both are in agreement that a certain level of resolution is required to give a result that is close for the converged solution. The current work indicates that this required resolution is dependent on choice of physics. The lead author does not see any reason why there might be a need to pick either numerics OR physics to "blame" for the difficulties, as the two are clearly related.*

*The lead author does not see how consideration of these two types of (lack of) convergence lends any clarity to the paper. However, since I believe that this paper and all the studies relevant to this paper deal with the former type (essentially convergent behaviour), I have tried to clarify what we mean by grid scale dependence without giving any mention of the second type grid scale dependence. Specifically we've modified the introduction and the first part of the results section.*

**2.**
I have no objection to picking a subset of the infinitely large number of possible sliding laws. However, what is missing is a discussion of why the authors think the various generalizations of Weertman sliding are more appropriate for more realistic ice sheet models. This is especially true when considering the simple assumption employed by the authors that effective pressure is a linear function of water depth **all the way to the ice divide**.

*This question is very similar to a question from the other reviewer, and we refer readers to our response to that review. However, we should acknowledge that Dr Bassis touches on relevant issues (see below). We have added a brief comment in the paper to indicate that the sliding relations in our paper – which are specifically intended as physically motivated modifications of Weertman sliding – do have origins in the quest for better representation of flow in real ice sheets. We also wish to clarify that we are not claiming that these specific sliding relations are generally more appropriate than other sliding laws, but rather that they are worth exploring, since arguments can be made both ways and we don't wish to see the whole community using a single sliding law.*

*Regarding the specific point about hydrologic assumptions and connectivity of all basal water to the ocean… actually we think arguments CAN be made for why this terrible assumption is actually better than assuming that basal drag is entirely independent of basal water, though clearly both assumptions are wrong. Traditional Weertman sliding is clearly wrong. Some of the other approaches and assumptions could be argued to be better, but none of them are great. And none of this is really the point of the current study. The point we want to make is that there is a relationship between model physics and resolution*

*requirements. We are not trying to say that everyone should use the same physics as we use in this study. Other researchers have to make their own decisions. But they may find it useful to think about basal physics of potential study areas and the implications for their study before picking a study area and before picking appropriate sliding laws and melt parameterisations. We do NOT want to advocate particular sliding laws or melt parameterisations, we just want to make the point that there are reasons why different forms of both may be relevant in different physical situations, and let others decide for themselves what to do with that information.*

*In short, we prefer not to enter into an argument about whether it is better to assume full hydrologic connectivity to the ocean or no connectivity at all. If reviewers/editor still think it essential then we will add such an argument, but we fear it could distract from the main point of the paper.*

*The importance of effective pressure in this study is near the grounding line. We think the reviewer agrees that the sub-glacial hydrologic system is more strongly connected to the ocean near the grounding line. Defining "near" in this context is both difficult and important, but well beyond the scope of this study.*

*The SR3 sliding relation can be regarded as progressively decreasing the influence of the ocean connection on effective pressure as the ice sheet becomes more firmly grounded inland.*

The fact that one sliding law is more convenient for numerical modelers doesn't necessarily make it the most physically appropriate sliding law.

*Exactly. We completely agree and we can't find anywhere in the paper where we claim the opposite. However, we have added text to the discussion to clarify that we do not advocate a particular sliding law or melt parameterisation.*

For instance, the 4 sliding laws predict very different ice sheet profiles with different divide thickness, slopes and even concavity of the profile. Can we say anything about the reasonableness of the sliding law assumptions based on observed ice sheet profiles (Pine Island?, Thwaites?, Lambert Basin, Siple Coast Ice Streams?). My guess is that one might find different sliding laws appear more appropriate for different portions of the ice sheet?

*This is exactly what motivated the researchers when modelling of the Antarctic ice sheet developed in the 1980's. We have added brief remarks about this to the paper. The concavity of the surface of fast flowing ice streams in West Antarctica highlights the deficiency of the Weertman relation (SR1) – ice stream velocities increase even though surface gradients and hence gravitational driving stresses are decreasing. The idea of a role for effective pressure (already introduced into sliding relations – e.g. by Lliboutry on theoretical grounds) and its identification with the "height about buoyancy" was considered a better representation (e.g. Bindschadler 1983) while Budd and his co-workers explored a variety of parametrisations to the limited data available. Dr Bassis also correctly observes that there are a variety of profiles associated with different fast flowing glacial systems. The Siple Coast ice streams*

*have very low profiles and a marked concavity compared to some relatively steep East Antarctic outlet glaciers.*

*In this paper our concern is with marine based ice sheets.*

**3.**
Selection of numerical parameter: Another question along a similar line of reasoning is how did the authors select the numerical "C" coefficients in the sliding laws?

*They were chosen to give approximately similar grounding line positions after the spin up. This results in very different thicknesses to the grounded ice sheet. We've added this comment to the methods section where the sliding relations are introduced.*
*Yes, the size of the drag coefficient can make a difference, as shown in Gladstone 2012 Annals paper, but by far the biggest difference is the change across the grounding line, and this is dominated by the choice of sliding relation. We've added a comment to this effect in the discussion.*

**4.**
Getting into the actual experiments, I'm puzzled by the fact that the authors spin up the models and find a steady-state that is independent of resolution, but then find that some of the forcing experiments do yield a dependence on grid resolution.

*This is only really puzzling if you expect a single steady state for a given forcing (see also our response to the other reviewer to a very similar question). In brief, some models tend to give better convergence in advance and some in retreat. This is likely due to choice of discretization and special treatments of the grounding line. Our model seems to give better results in advance than in retreat. It is plausible that a change in the way forcing is applied could cause this to change, and indeed in the melt experiments our model does better in retreat than advance. This is already mentioned in the results section.*

**5.**
Similarly, I would urge the authors to dig a bit deeper into the physics behind the results.

*Actually, resolving the transition zone adequately is only part of the problem, possibly less than half the problem. There is a discrete feedback between the forcing and the state of the system. In a continuous system, advance of the grounding line is incremental with forcing. You can have a tiny change in forcing and this can cause a tiny change in grounding line position. This in itself results in a tiny change in forcing because of the increase in total basal drag (if we're talking about grounding line advance), which in itself causes further advance. Hence there is a positive feedback here between state and forcing. In the model a forcing perturbation of a certain size is needed in order to cause the grounding line to advance by one element or grid cell. A very small change in forcing SHOULD cause an increase in total basal drag because it SHOULD make the grounding line advance a bit. But it doesn't because the grounding line hasn't moved at all. This discretisation of what should be a*

*continuous positive feedback between model state and model forcing causes artificial stickiness in the model which is, to my mind, at the heart of the resolution problems. However, it is very difficult to separate this argument out from the simple issue of having sufficient resolution to resolve the transition zone itself, and I fear presenting this argument in its current form may not be helpful. We could put this in if reviewer and editor feel it adds value to the study.*

*We've added some discussion to the discussion about the relevance of our study to using a real hydrology model.*

**Comments on presentation**
The writing in the manuscript is generally clear, but stylistically, the manuscript contains many paragraphs that consist of one or two sentences. The abstract, for example, contains 4 distinct paragraphs! Normally, abstracts are a single paragraph. Traditionally, at least in the US, paragraphs are units that contain coherent ideas. Paragraphs depend on each other, but paragraphs themselves should be coherent. I found it jarring to have indentation that has little relation to the organization of ideas. For example, on Page 4 line 9 contains a paragraph with the single sentence starting with "This parameterisation is similar to that used in the . . ." The next paragraph begins with a definition of a variable. Neither of these are coherent or can stand on their own. This peculiar isolation of sentences into incoherent paragraphs was jarring to my American sensibilities, but this is perhaps a European or Cryosphere Discussion stylistic preference. I leave it at the editors and/or authors discretion as to whether this is stylistically appropriate.

*We aim for our manuscript to be readable to all, even Americans. We're happy with the current multi-paragraph abstract (but willing to merge all the paragraphs into one if the reviewer and editor still feel strongly about this), but we've merged paragraphs in several other places, including the specific example mentioned by Dr Bassis. From a purely aesthetic point of view, note that the paragraphs will look longer when in double column format…*

I was also flummoxed by the bestiary of acronyms. I was confronted with ISM, MIP, MIS all in the first several paragraphs and later hard to deconstruct SR1 and SR2, ALMW, ALMN amongst what seemed like many others. I have strong personal preferences to avoid all unnecessary acronyms and, unless the authors are forced to pay by the word, I urge the authors to use complete words instead of acronyms whenever possible. If this is inconvenient or impossible, then I would suggest a table that readers can easily refer to that provides us with a dictionary of acronyms that we can conveniently look up.

*Surely ISM, MIS and MIP are very commonly used abbreviations? Also, they are all introduced together in the first paragraph, which is pretty convenient. Do we need a table for just these three? Or should we spell them out everywhere? We don't have a fundamental objection to either but don't fully understand the need. Does the editor have a recommendation here?*

*Some of the experiment names were indeed rather beastly. We've renamed most of these, mostly with shorter abbreviations now, and capitalised the relevant letters in their description in the table, so hopefully now the experiment names will make more sense. After revision we more seldom insert acronyms beyond the experiment labels, and we tried to accompany these by reminders which simulations we are discussing.*

My last comments on presentation concerns the figures, which although physically appealing, were difficult for me to parse. For example, I struggled to understand Figure 1. According to the caption, panels show the steady and final states for the basal melting experiment. However, the top panel doesn't have the gray shaded profile. Why not? Moreover, given that the shaded gray profiles obscure parts of the initial profile, I would prefer to see this figure broken into two figures. The first would show the initial, post spin-up profiles. The second would show the final, post simulation profiles (with velocities). This would allow readers to more easily be able to see the differences in the shapes of the initial and final profiles and contemplate the role that the sliding law in modifying the morphology of the profiles and the transition zone. This figure also needs a scale bar to show horizontal distance (and possibly vertical elevation).

For Figure 2, why are only 2 steady states shown in Figure 2? Why not show all of them (as in Figure 1)?

*Note that the top panel of Fig 1 has no grey profile because (as mentioned in the text) SR1 was not run in retreat due to melting. Fig 1 has been modified so that the grey profiles are now only outlines. Given that Fig 1 shows the whole domain for all four sliding relations, distance labels make the plot cluttered and we feel are not helpful. However, these could still be added if the editor and reviewers still think they are needed.*

*Fig 2 is intended to give a more detailed view of the two types of stress regime. The other simulations are not qualitatively different from the two shown. We prefer to show just what we feel supports our key points. However, as above, if the editor and reviewers still feel showing all four would be beneficial this could be done.*

Figure 3 was difficult for me to parse. I would urge the authors to considering using color schemes that are more appropriate for the color blind and/or line types that don't depend as much on color.

*We have reordered the legend to indicate the general progression of curves down the plot, and drawn attention to this in the caption. An alternative and easier to view colour scheme was not obvious.*

**Typographical comments:**
page 1 near line 10: even with –> even with

*You mean page 2? Changed, thanks*

page 3 above line 10: incomplete sentence "These values for p and q are chosen for

simplicity, and deviate" ????

*Thanks, completed sentence.*

From a physical perspective, Equation 2 is problematic. It assumes that the ice sheet has a basal hydrology sentence that is perfectly connected with the ocean all the way to the ice divide.

*A basal hydrology scheme?  Yes, this is physically problematic.  So is the assumption that basal water pressure is constant all the way to the ice divide, which is implied in SR1.  We don't think it is obvious which is worse (though we would argue the latter is worse), but we're not going to resolve this question here.  If it were obviously worse than the alternative then we would agree that it would need some justification.*
*As already indicated SR3 can be regarded as representing a modification which effectively increases effective pressure faster inland, in as much as it approaches the Weertman result for firmly grounded ice.*

The constant "C" in Equations (3)-(6) has different units in the suite of basal drag parameterizations. To avoid confusion, I recommend using a different symbol or appending subscripts to more clearly indicate that the numerical value of C (and its units) are not identical in all experiments.

*They have subscripts now.*

Line 4, line 15 and elsewhere: need a space between the number and unit "100m" –> "100 m"
After section 2.3 "1800km" –> "1800 km"

*Hopefully we've caught all these now, thanks.*

Accumulation rate: What is the motivation for the accumulation rate defined by Equation (11). This appears to predict linearly increasing accumulation with zero accumulation at the ice divide? Why not use a constant accumulation, which would seem to be more realistic for much of Antarctica?

*It makes the ice divide boundary condition easier and it is more like the real Antarctica.  Ok, linearity is not correct, but there is certainly more accumulation near the margins than in the interior.*

Spin-up is independent of resolution. This seems to indicate that resolution only matters in some experiment types???

*This is discussed above already.*

Page 6, line 6 "focusses" –> "focuses"

*Fixed, thanks!*

Page 6, line 7 missing commas –> "The spinup simulations do however vary" –>The spinup simulations do, however, vary"

*Fixed, thanks!*

Page 6, line 21: "But even SR4b still shows significant resolution dependency." –> sentence fragment, consider revising.

*Revised!*

Page line 23: "Since" indicates time (e.g., I haven't slept since yesterday). I think the authors want "Because" (Because SR1 and SR4a . . .)

*Yes, that is probably better, revised.*

Page 7, line 4 missing space: "20ka" –> "20 ka"

*Fixed, thanks.*

Page 7, line 29, "Any resolution dependence in a model is inevitably non-physical. Ideally model behavior should converge with finer resolution." This line is perplexing and indicates my fundamental misconception. Are the authors arguing that grounding zone position fails to converge with increasing resolution and is, hence, resolution dependent or are they arguing that grounding line position requires finer resolution than they have available. There is no particular guarantee that a numerical model with a priori specified resolution will be an accurate representation of a given set of partial differential equations. This is especially the case when the resolution of the model is more coarse than the fundamental scale of the system.

*At this particular point we're not arguing either, rather we're discussing desirable properties of a modelled system. At no point do we attempt to argue for failure to converge in general, though we do not go to sufficiently fine resolution in some cases to demonstrate convergence. The main point of our paper is the exploration of situations where convergence – in our rather operational sense – is achieved with relatively coarse resolution, compared to previous studies – that typically use SR1. The cases that don't converge across the range of our coarse resolutions simply identify conditions that are not amenable to the coarser resolution modelling.*
*This particular couple of sentences are not essential and we've simply removed them to avoid confusion.*

Page 8, Line 10-15: I don't think it is true that basal melting \*requires\* subglacial discharge. Because the ice near the grounding line is (usually) located at a depth that is much greater than the melting point of sea water, one expects melt near the grounding line of ice shelves with deep grounding lines, even in cold cavity ice shelves. In the traditional ice-pump theory, the cold melt water mixes with sea water and forms a buoyant plume. I'm puzzled by the argument that subglacial discharge is required to initiate the process? Are the

authors arguing for massive super cooling near grounding lines? Is this usually observed? Models of submarine melt under ice shelves rarely include grounding discharge and yet predict realistic patterns of basal melt. Why is this if submarine discharge is a crucial component of the process? It also seems like the relevant length scale over which basal melt must go to zero is going to related to the characteristic width of the buoyant plume. Can this be estimated and used to better constrain the parameterization of basal melt?

*Having re-read the original text we do not see where in these lines we claim that sub glacial outflow is required in order to get basal melting. It is a factor that influences basal melting close to the grounding line, which is important. We are not arguing for massive super cooling near grounding lines. We've modified the wording in this section. Hopefully it will be less prone to misinterpretation, but we're not very confident about this as we can't see where in the original text we claimed that subglacial outflow was strictly required.*
*We have revised the discussion of subglacial meltwater input, and clarified the observational point that there are high basal melt rates are often inferred near deep grounding lines (Rignot and Jacobs 2002). We have also included reference to Jenkins (2011) which explores the capability of subglacial melt at the grounding line to generate high melt rates close to the grounding line. Models of submarine ice melt are increasingly considering subglacial melt water discharge at the grounding line.*

---

## Author Response (AR1)

The author's response is given in the two individual replies to the two reviewers. A manuscript highlighting all changes compared to the original submission as attached as a supplement (diff.pdf).

[revised manuscript text omitted]